# An Experimental Study of Zinc Evaporation from Bottom Zinc Dross at Atmospheric Pressure and in Inert Atmosphere with Integrated CFD Modelling

**DOI:** 10.3390/ma17184627

**Published:** 2024-09-20

**Authors:** Katarína Pauerová, Róbert Dzurňák, Jarmila Trpčevská, Pavol Liptai, Tomáš Vindt

**Affiliations:** 1Faculty of Materials, Metallurgy and Recycling, Institute of Recycling Technologies, Technical University of Kosice, Letna 9, 042 00 Kosice, Slovakia; jarmila.trpcevska@tuke.sk (J.T.); pavol.liptai@tuke.sk (P.L.); tomas.vindt@tuke.sk (T.V.); 2Faculty of Materials, Metallurgy and Recycling, Institute of Metallurgy, Technical University of Kosice, Letna 9, 042 00 Kosice, Slovakia; robert.dzurnak@tuke.sk

**Keywords:** bottom zinc dross, zinc recycling, high-added-value products

## Abstract

In the present study, the recycling process of bottom zinc dross was performed by evaporation and subsequent condensation at 800 °C for 30 min with an observed argon flow rate of 100–400 mL/min to ensure an inert atmosphere, to observe the evaporation rate and final form of the product. Under the set conditions of over 98% zinc purity, products in the form of nanofibres (thickness 500 nm), powder (size of spherical particles 2–5 μm), dendrites, and metallic forms were obtained. The employed mathematical modelling (via Ansys 2023R1 software) predicted the behaviour of the argon flow current in the quartz tube, as well as the temperature gradient in the quartz tube and in the close vicinity of the zinc sample. Via Inventor 2014 software, the rate of zinc sample heating was calculated. All the simulations were compared with the physical measurements and correlation was proven.

## 1. Introduction

Bottom zinc dross originates from the hot dip galvanizing process. This waste contains over 96% zinc and the main impurity iron, which makes it impossible to reuse this waste in the galvanizing process. The presence of iron in molten zinc causes several problems, such as an increase in the melting temperature of zinc or a deterioration in the surface quality of galvanized parts. Currently, no refining process applies, and the bottom zinc dross is sold for chemical compound production, mostly ZnO. However, many studies have been carried out with the aim of refining zinc from this type of waste. Since the boiling and melting points of zinc are relatively low, heat treatment may be a suitable method for refining zinc-bearing waste. In particular, the evaporation method can be implemented. Several approaches have been considered when investigating zinc evaporation. Zinc can evaporate at a normal pressure (1 atm), and at lower pressures down to vacuum. The well-known high affinity of zinc for oxygen requires the provision of an inert atmosphere. 

In his work [1], Yu Ming Su investigated the evaporation of zinc at atmospheric pressure. The author observed the rate of evaporation of zinc at different temperatures in a single atmosphere of argon, carbon monoxide, and hydrogen. Ingots of high-grade zinc, containing 0.0002% Fe, 0.0019% Pb, and 0.0017% Cd, were used. The flow rate of gas was adjusted to give a flow rate of 10 mL/min. Th studied temperatures ranged from 650 °C to 850 °C. The evaporation surface of the sample was 35 cm^2^. The rate of evaporation was found to increase slowly at lower temperatures and rapidly at higher temperatures. As assumed, the highest zinc evaporation rate was reached within the highest temperature (850 °C). Regarding the use of gas and its effect on the zinc evaporation, the evaporation in hydrogen was twice the rate of argon. The evaporation rate in the argon–oxygen atmosphere was independent of the oxygen concentrations of the mixtures studied. The presence of oxygen caused the evaporation rate to be lower than that in pure argon. The carbon monoxide evaporation rate was approximately the same as that of argon. This study did not focus on the form or purity of the final product.

The evaporation of zinc in an argon–hydrogen mixture was evaluated in the work of Matula [2]. The hydrogen content in the mixture varied in the amounts of 5, 15, and 35% at the temperature of 675 °C. The experiments were conducted for 30 min. Based on the results, it was shown that with the increased hydrogen content, the rate of zinc evaporation increased. The authors of [3] investigated the effect of helium on the rate of evaporation of zinc from brass scraps under atmosphere pressure. Brass scraps containing 10.53 wt% Zn were used for the experiments. The experiments were conducted using the thermogravimetric method at 1080 ÷ 1240 °C in a helium atmosphere. The levels of zinc removal from copper ranged between 82% and 99%. The values of the overall mass transfer coefficient for zinc k_Zn_, determined based on the experimental data, ranged from 4.74 to 8.46 × 10^−5^ ms^−1^. Labaj [4] carried out an investigation of zinc evaporation possibilities in a helium and carbon monoxide atmosphere with a gas flow of 50 L/min at the temperatures of 675, 700, 725, 750, and 775 °C. The samples were heating at a given temperature for a period of 30 min. On the basis of research, it was shown that the rate of zinc evaporation in a helium atmosphere was greater than the evaporation rate under the atmosphere of carbon monoxide. Xiang [5] made another comprehensive study of zinc evaporation within the temperature range of 650 °C to 800 °C and in an inert nitrogen atmosphere. Zn–Mn batteries were used for experiments. The nitrogen pressure was set up in the range from 1 Pa to 10,000 Pa. With an increasing temperature, the zinc evaporation rate also increased. Increasing the pressure in the system led to the formation of zinc nanoparticles. A high separation efficiency of 99.68% was achieved when the heating temperature rose to 1073 K under 10,000 Pa of nitrogen gas pressure. Qi et al. [6] studied the vacuum evaporation of zinc in the production of high-purity zinc. The sample chosen for the experiments contained 7.62% Pb, 0.08% Cd, 3.77% Fe, 0.28% Cu, 0.01% Al, and 0.02% Sn. The evaporation area was 7.07 cm^2^. The evaporation rates were assessed based on the observed pressures ranging from 5 to 400 Pa at the temperatures of 400, 450, 500, and 550 °C. High-purity zinc was obtained at a system temperature and pressure of 450 °C and 5 Pa. The concentration of Pb (0.25%) in the experimental product was low. 

In their experimental study, Zhang et al. [7] investigated the volatilisation and condensation of zinc at pressures of 10 Pa and 200 Pa. A sample of 99.9% zinc was used for the research. The argon flow rate was set at 50 mL/min. The author compared the final form or structure of the condensed zinc within these pressures. It was found that at a pressure of 10 Pa, the zinc had no liquid phase, and the zinc was condensed directly by desublimation. At a pressure of 200 Pa, the zinc completely grew in the form of molten droplets above 420 °C, and dendrites appeared near the melting point. Research by Smalcerz et al. [8] has been published regarding zinc evaporation from aluminium alloys, with the content of zinc in the alloy being 6.3%. Zinc was evaporated at pressures ranging from 1000 Pa to 10 Pa and in a temperature range of 680 °C to 830 °C. The best results (96% zinc efficiency) were obtained with the reduction in the operating pressure to 10 Pa and an increase in temperature to 830 °C. Barakat M.A., in his study [9], investigated zinc refinement with the addition of aluminium agents in a temperature range of 600 °C to 900 °C. A refined zinc was obtained by adding 0.4% Al at 700 °C after 2 h of refining while lowering the iron concentration from 3.2% in the hard zinc to 0.5% in the refined product. 

As mentioned above, the refinement of zinc from zinc-containing wastes, or from artificial samples, was investigated in various studies. However, there is a lack of studies on the evaporation of zinc from a waste sample in an inert atmosphere with a flow rate applied to monitor the evaporation efficiency and the effect of the flow rate on the final form of the product. This paper deals with this issue comprehensively and, therefore, the simulation of a particular phenomenon was evaluated using Ansys software to predict the behaviour of the argon flow rate in the quartz tube and in the vicinity of the zinc sample during the process of evaporation.

Computational Fluid Dynamics [10,11,12] (CFD) is a key tool in scientific and industrial research for simulating complex processes like chemical kinetics [13], flow and turbulence [14], and heat transfer [15]. It combines mathematical modelling with experimental data to replicate real-world behaviour, using software like Fluent, CFX, and COMSOL. CFD solves core differential equations, such as the Navier–Stokes and Fourier–Kirchhoff equations, to analyse detailed processes across various fields.

In one study in the field of zinc processing, CFD modelling has been used to analyse the synthesis of ZnO vapour from zinc metal [16]. The study addressed the chemical kinetics of the oxidation process of zinc metal with a description of the definition of the boundary conditions in the mathematical model. The CFD modelling resulted in examples of ZnO mass concentration field distributions and temperature profiles. Huda [17] modelled the processing of zinc steres as part of his scientific work. Schwarz [18], in his work, analysed the possibilities of using CFD models to improve zinc processing.

In their study [19], Rybdylova et al. built a model for multi-component droplet heating and evaporation, and presented its implementation into Ansys Fluent. The model was applied to the analysis of acetone/ethanol droplet heating/cooling and evaporation. The predictions of the customized version of Ansys Fluent, with the new model implemented into it, were verified against the results predicted by the previously developed in-house code based on the analytical solutions to the heat transfer and mass diffusion equations. The agreements between the predictions of those codes were proven to be very good for multi-component droplets comprising acetone with 25, 50, and 75% ethanol, with input parameters comparable with those used for the previous verification and validation of the in-house code. A model for the heating and evaporation of droplet clouds and its implementation into Ansys Fluent were investigated in [20]. In that work, the mathematical models of the gas–droplet flow and their implementation into Ansys Fluent were described. The testing of the new model was based on the analysis of a droplet cloud in a steady back-step flow. The temperature gradients in the flow were shown to have a significant indirect effect on the evolution of the droplet number densities via the effects of these gradients on the flow velocity. The authors of [21] studied the performance of the porous plate water sublimator numerically and experimentally using the CFD model. The proposed simulation could track the evaporation and sublimation process reasonably before reaching the lowest temperature. Based on the verified model, the sublimator was predicted under different feedwater flow rates, heating fluxes, and initial locations of the liquid–gas interface. 

In this work, the prediction of heat transfer through the zinc sample, the time required to deflate the system, the effect of the gas flow on the temperature in the vicinity of the zinc sample, and the possible turbulent behaviour of the gas flow were investigated. These models served as supporting data to better understand the behaviour of a zinc sample placed in a quartz tube during the evaporation process. These were not direct models of the evaporation process itself. The overall zinc evaporation process with subsequent condensation has not been modelled yet, as it is a very complex simulation, and it would have required very challenging/demanding mathematical calculations. 

## 2. Materials and Methods

### 2.1. Materials

Zinc-bearing waste samples (Figure 1b) with dimensions of 25 × 10 × 5 mm were used for all the experiments and simulations. Samples were analysed to determine contents of zinc and other elements (Table 1) by AAS (atomic absorption spectrometry analysis) using model ContrAA 700 (Analytik Jena, Jena, Germany). The range of the composition was averaged from 10 samples taken of various locations from the bottom zinc dross provided. Phase analysis was realized by XRD (X-ray diffraction) using model XPERT PRO RV-11/2010, PANalytical, Worcestershire, UK, as shown in Figure 2 and Table 2.

According to the XRD analysis in Figure 2 and Table 2, the bottom zinc dross consists of metallic zinc and impurities, such as metallic lead and intermetallic compound FeZn_13_. 

### 2.2. Experimental Apparatus and Conditions of Evaporation Process

The samples were evaporated in an electric resistance furnace, which allows temperatures up to 1100 °C. The samples were inserted into an 18-millimeter-internal-diameter quartz tube of varying lengths. The process temperature was set at 800 °C. The apparatus used for the experiments can be seen in Figure 3. A view inside the furnace with a zinc placement and location of a thermocouple is shown in Figure 4. The entire system was blown with an inert argon gas of the highest purity of 99.99% to prevent oxidation of the zinc vapours. The observed time of the evaporation process was 30 min. Investigated argon flow rates were 50, 100, 200, 300, and 400 mL/min. The argon flow rates were chosen based on the intensity of the current within chosen quartz tube diameter. The flow rate under the 50 mL/min was too slow to carry the zinc vapour into the condensation zone, and the rate over 400 mL/min was too fast to observe and evaluate the condensed products effectively. 

Samples were weighed before and after evaporation process, and the weight difference was noted. The mass loss after evaporation within the set argon flow was evaluated. Mathematical model simulations of several phenomena were performed for better assumptions of the process setup (described in the next section).

**Figure 3 materials-17-04627-f003:**
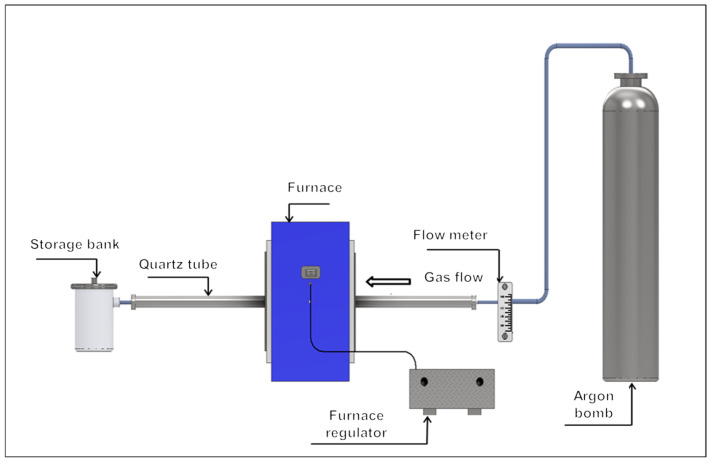
Furnace used for zinc evaporation (modelling via Inventor).

**Figure 4 materials-17-04627-f004:**
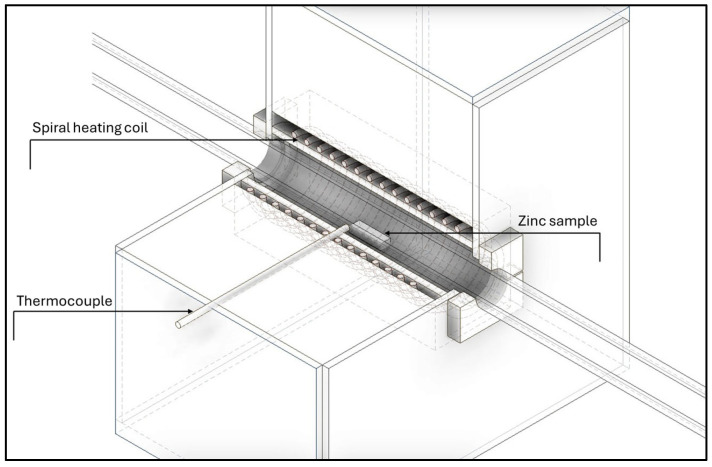
A view inside the furnace unit.

When measuring the temperature gradient in the quartz tube, the argon flow rates were 100 and 400 mL/min, respectively (discussed in the Section 3.3) PTTK-TKb-60-2-SP (NiCr-Ni) thermocouple (Meratex, s.r.o., Kosice, Slovakia) was used, for which the manufacturer states a measurement uncertainty at the level of 2.5 °C within the temperature range up to 1200 °C [23]. Thermocouple was placed at the bottom and in the centre of the quartz tube’s cross-section. 

### 2.3. Modelling of the Chosen Phenomena Using Ansys Fluent and Inventor

The location of the sample in the quartz pipe can be seen in Figure 5. In geometry, the entire quartz tube was simulated. The length of the quartz tube was 400 mm, and inner and outer diameter were 18 mm and 20 mm, respectively. 

Solving the Navier–Stokes equations using CFD modelling requires that the geometry of the model in which the differential equations are computed is discretized into a computational mesh. The proposed computational mesh is formed by tetrahedral cells (four-edge), and in the wall region it is formed by prismatic cells (six-edge). The densification in the wall region is important in terms of modelling boundary layers, within which a rapid change in the velocity field near the walls occurs. The quality of the computational mesh affects the quality of the computation, and also the computational load on the workstation. The quality of the network was monitored by cell skewness (Skewness). The worst cell quality was 0.69, which, based on Table 3, means that it was within the range defined for a good network. The number of cells in the computational network was 448,000. The computational network is shown in Figure 6. A mesh independence analysis was conducted for 200,000, 300,000, and 400,000 cells at argon flow rate of 100 mL/min within a turbulent mode. The result of this analysis is shown in Figure 7. 

The analysis showed that the network with 400 k cells performed the best. The difference in the endpoints within FM (physical measurement) and all MM (mathematical simulation) was due to the actual furnace design. When the furnace was modelled in Ansys, the resistant linings were not modelled.

The character of the flow defined by the Reynolds criterion was laminar within the design flows (Table 4). The k-ε model was designed for turbulent flow, but due to the diffusion of zinc vapours into the argon stream, this model was preferred over models designed for laminar flows. The turbulent model proved to have a better correlation within higher argon flow rates when compared with physical measurement. Verification of the results obtained from mathematical modelling with physical measurement is discussed in Section 3. 

The Reynolds-averaged Navier–Stokes (RANS) k-ε model was used in the mathematical model, which is one of the most widely used models in practice. The model solved for turbulence based on Equations (1) and (2). The kinetic energy of turbulence k with kinetic energy dissipation ε was obtained from the following transport equations [24]: (1)$$\frac{\partial }{\partial t}\left(\rho k\right)+\frac{\partial }{\partial xj}\left(\rho ku_{j}\right)=\frac{\partial }{\partial x_{j}}\left[\left(\mu +\frac{\mu_{t}}{\sigma_{k}}\right)\frac{\partial k}{\partial x_{j}}\right]+G_{k}+G_{b}-\rho \varepsilon -Y_{M}+S_{k}$$
(2)$$\frac{\partial }{\partial t}\left(\rho \varepsilon \right)+\frac{\partial }{\partial xj}\left(\rho \varepsilon u_{j}\right)=\frac{\partial }{\partial x_{j}}\left[\left(\mu +\frac{\mu_{t}}{\sigma_{\varepsilon }}\right)\frac{\partial \varepsilon }{\partial x_{j}}\right]+\rho C_{1}S_{\varepsilon }-\rho C_{2}\frac{\varepsilon^{2}}{k+\sqrt{v\varepsilon }}+C_{1\varepsilon }\frac{\varepsilon }{k}C_{3\varepsilon }G_{b}+S_{\varepsilon }$$
where the following apply:G_k_—generation of turbulence kinetic energy due to velocity gradients;G_b_—generation of kinetic energy of turbulence due to buoyancy;Y_M_—contribution from fluctuating dilatations in compressible turbulent flow to total dissipation;C_1ε_ C_2_,C_3_—model constants;σ_k_, σ_ε_—model constants—turbulent Prandtl numbers for k and ε (-);S_k_, S_ε_—user-defined source members.

The general form of the mass (3), momentum (4), and energy (5) conservation equation solved by Ansys 2023 R1 software is [24]
(3)$$\frac{\partial \rho }{\partial t}+\nabla \cdot \left(\rho \vec{v}\right)=0$$
where the following apply:Ρ is the fluid density;T is time;$\vec{v}$ is the velocity vector of the fluid.∇⋅(ρ$\vec{v}$) represents the divergence of the mass flux (rate of mass flow per unit area).

The general form of the momentum conservation equation in the i-th direction is
(4)$$\frac{\partial }{\partial t}\left(\rho v_{i}\right)+\nabla \cdot \left(\rho \vec{v}v_{i}\right)=-\frac{\partial p}{\partial x_{i}}+\nabla \cdot \tau_{ij}+\rho g_{i}+F_{i}$$
where the following apply:$v_{i}$ is the velocity component in the iii-th direction;P is the pressure;$\tau_{ij}$ is the viscous stress tensor, representing the viscous forces acting on the fluid;$g_{i}$ represents the gravitational acceleration in the i-th direction;$F_{i}$—is an external body force (e.g., due to electromagnetic fields or other forces).
(5)$$\frac{\partial }{\partial t}\left(\rho E\right)+\nabla \cdot \left(\vec{v}\left(\rho E+p\right)\right)=\nabla \cdot \left(k\nabla T\right)+S_{E}$$
where the following apply:E is the total energy per unit mass, which includes internal energy and kinetic energy;T is the temperature;K is the thermal conductivity of the fluid;∇⋅(k∇T) represents the heat conduction (Fourier’s law);$S_{E}$ represents energy sources (e.g., due to chemical reactions, radiation, or other heat sources).

Model constants:C2-Epsilon—1.9TKE Prandtl number—1TDR Prandtl number—1.2Energy Prandtl number—0.85Wall Prandtl number—0.85Turbulent Schmidt number—0.7

Species transport was chosen with mixture-template and volume-weighted-mixing-law option for density within the elements argon and zinc vapour. Boundary conditions for inlet were chosen as follows: mass flow inlet, mass flow rates 1.487 × 10^−6^, 2.9736 × 10^−6^, 5.946 × 10^−6^, 8.919 × 10^−6^, and 1.189 × 10^−5^ kg/s for the argon flow rates of 50, 100, 200, 300, and 400 mL/min (explained in the Table 4). Thermal—total temperature 300 K, Species—Species mass fractions for argon with value 1 (with same setup for each flow rate). Outlet was set for outflow with flow rate weighting value 1 (with same setup for each flow rate). Wall conditions were as follows: stationary wall, standard roughness models (default setup), temperature set to 1073 K (with the same setup for each flow rate). Number of time steps was 1800 with time step size 1 s, max iterations/time step 20. The convergence criteria were set as default value given by software. 

Regarding the measurement of the heat transfer trough the sample using Inventor software, the temperature change in the modelled sample was measured generally in a whole object, neglecting the flow intensity.

#### 2.3.1. Measured Phenomena

A zinc sample was drilled with a hole (Figure 8a) and inserted with thermocouple through the hole in the quartz tube (Figure 8b) to measure the change in temperature in the sample within the time until the point at which the temperature of 800 °C was reached. 

#### 2.3.2. The Character of the Gas Flow in the Tube

The character of the gas flow in the tube was predicted by calculating the Reynolds number according to Equations (6)–(8), with 4.65 for 100 mL/min and 18.61 for 400 mL/min of Ar flow (resulting from the Table 4).
materials-17-04627-t004_Table 4Table 4Values/calculations required for simulations and determination of the character of the flow at the condition: d tube—0.018 m, S tube—2.54 × 10^−4^ m^2^, ρAr—1.7838 kg/m^3^ at 25 °C, ρAr—0.4538 kg/m^3^ at 800 °C.Q_Ar_mL/min50100200300400500mL/s8.33 × 10^−1^1.673.3356.678.33m^3^/s8.33 × 10^−7^1.67 × 10^−6^3.33 × 10^−6^5.00 × 10^−6^6.67 × 10^−6^8.33 × 10^−6^m_Ar_kg/s1.487 × 10^−6^2.973 × 10^−6^5.946 × 10^−6^8.919 × 10^−6^1.189 × 10^−5^1.4865 × 10^−5^w_25 °C_m/s3.27 × 10^−3^6.55 × 10^−3^1.31 × 10^−2^1.96 × 10^−2^2.62 × 10^−2^3.27 × 10^−2^w_800 °C_m/s1.61 × 10^−2^3.23 × 10^−2^6.46 × 10^−2^9.69 × 10^−2^0.130.16η_Ar_ 25 °Cη_Ar_ 800 °CPa.s2.26 × 10^−5^2.26 × 10^−5^2.26 × 10^−5^2.26 × 10^−5^2.26 × 10^−5^2.26 × 10^−5^5.63 × 10^−5^5.63 × 10^−5^5.63 × 10^−5^5.63 × 10^−5^5.63 × 10^−5^5.63 × 10^−5^ν_A_ 25 °Cν_A_ 800 °Cm^2^/s1.27 × 10^−5^1.27 × 10^−5^1.27 × 10^−5^1.27 × 10^−5^1.27 × 10^−5^1.27 × 10^−5^1.24 × 10^−4^1.24 × 10^−4^1.24 × 10^−4^1.24 × 10^−4^1.24 × 10^−4^1.24 × 10^−4^Re 25 °CRe 800 °C^−^4.659.3118.6127.9237.2246.532.344.689.3614.0418.7323.41
(6)$$Re=\frac{w\cdot d}{\nu_{Ar}}$$
(7)$$\nu_{Ar}=\frac{\eta Ar}{\rho AR}$$
(8)$$w_{Ar800\;{}^{\circ}C}=w_{Ar25\;{}^{\circ}C}\times \left(1+\frac{T_{1073}}{T_{298}}\right)$$
where the following apply:Q_Ar_—volumetric argon flow in m^3^·s^−1^; w—rate of argon flow in m·s^−1^; d—diameter of the quartz pipe in m; ν_Ar_—kinetic viscosity in m^2^·s^−1^;η_Ar_—2.096 × 10^−5^ Pa·s;ρ_AR_—density of argon at 25 °C (1.7838 kg/m^3^), at 800 °C (0.4538 kg/m^3^).

The dynamic viscosity of argon as a function of temperature is shown in Table 5 and in Figure 9.

From the equation in the graph in Figure 9, the relationship can be deduced as follows:(9)$$y=a+b\times T+c\times T^{2}$$
specifically,
(10)$$y=2.1129+0.0062\times T-0.000002\times T^{2}$$

From the relationship, the value of the dynamic viscosity of argon at 800 °C can be calculated to be 5.63 × 10^−5^ Pa.s.

## 3. Results and Discussion

Using Ansys Fluent (version 2023 R1, supplier TechSoft s.r.o., Poprad, Slovakia), a model of the time required to expel air from the system (Figure 10), the change in temperature within chosen flow rates (Figure 11), heat transfer trough the sample (Figure 12), and the character of the flow rate in the quartz tube (Figure 13) were modelled.

Due to the comprehensive study carried out, this section is divided into several individual parts for better clarity of the results.

As can be seen from Figure 10, under the specified conditions, all the air was purged from the system within 100 s. The boundary condition at the input of the mathematical model was defined by the mass flow rate of the argon, which corresponded to the experimental measurements. The argon flow rate for this particular demonstration was set at 2.973 × 10^−6^ kg·s^−1^ (100 mL/min—Table 4).

**Figure 10 materials-17-04627-f010:**
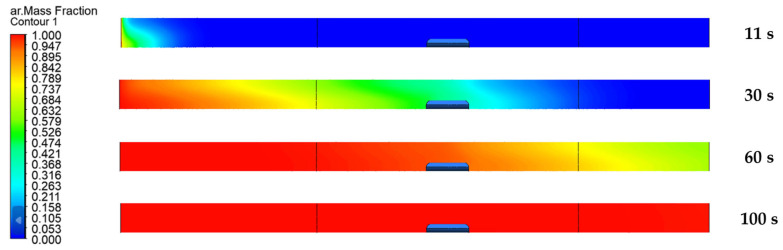
Model of deflation within argon flow rate of 100 mL/min.

### 3.1. Effect of the Argon Flow Rate on the Temperature in the Quartz Tube

The effect of the argon flow rate on the temperature in the quartz tube and the temperature in the close vicinity of the zinc sample itself was observed using Ansys software (Figure 11). 

**Figure 11 materials-17-04627-f011:**
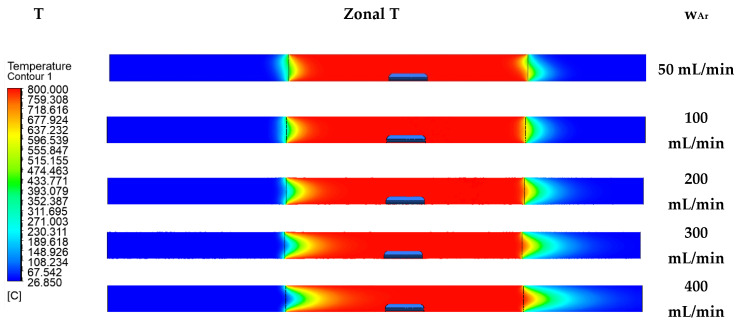
Argon flow rate impact on the temperature.

Generally, the argon flow rate influences the overall sample bypassing and, thus, the kinetics of the zinc evaporation. Based on this particular software prediction, a possible risk of the uneven heating of the sample could be inferred. Comparing the lowest and highest argon flow rates from Figure 11, a temperature drop can be observed in the inlet zone of the quartz tube placed in the furnace. The temperature drop can be observed in the first third of the quartz tube. However, this range of the flow rate does not negatively affect the observed temperature in all the sample surroundings. If the temperature in a portion of the sample were to be lowered by the argon flow, this would cause a change in the rate of zinc evaporation from the sample in the affected portion of the sample. The reason for the temperature drop within the increased argon flow rate is the ambient temperature of the injected gas. The higher the flow rate with the ambient temperature, the shorter the heating time of the gas. An evaluation of the measured data (Figure 11) through simulations with real measurements in the laboratory is presented in Section 3.3.

### 3.2. Heat Transfer through the Sample

Another observed phenomenon was the heat transfer trough the zinc sample within time using Inventor software. The simulated model matched the experimental conditions. Regarding the simulation (Figure 12), it was found that during 210 s of sample heating in the furnace under the 800 °C, the sample reached the temperature of 778 °C. A physical measurement was carried out to determine the rate of heat transfer through the object/sample of zinc placed in the furnace. A hole approximately 3 mm in diameter was drilled into the zinc sample (Figure 8a). A NiCr-Ni thermocouple was used to measure the temperature change and inserted into the hole in the sample (Figure 8b). An evaluation of the measured data with real measurements in the laboratory is presented in Section 3.3.

**Figure 12 materials-17-04627-f012:**
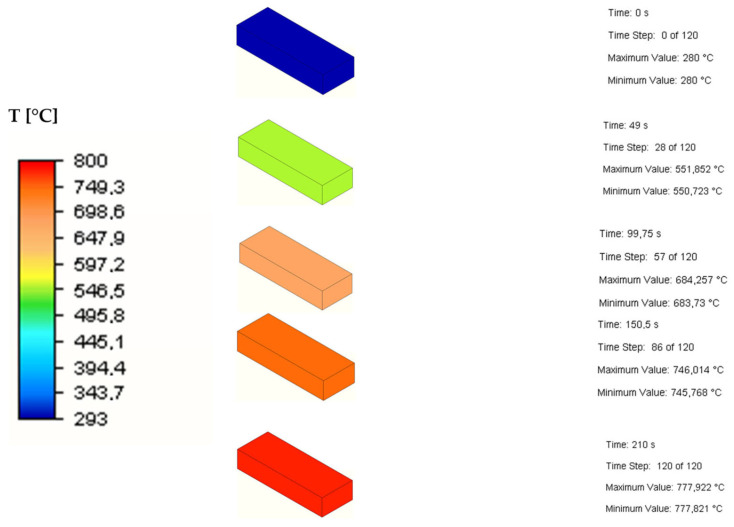
Heating of the sample over time.

Based on the Reynolds number, the character of the gas flow was laminar. Despite the laminar type of determination, turbulent condition was observed in the tube at lower flow rates, as can be seen in Figure 13. This software prediction was also confirmed by the physical observation of the evaporation under the laboratory conditions.

**Figure 13 materials-17-04627-f013:**
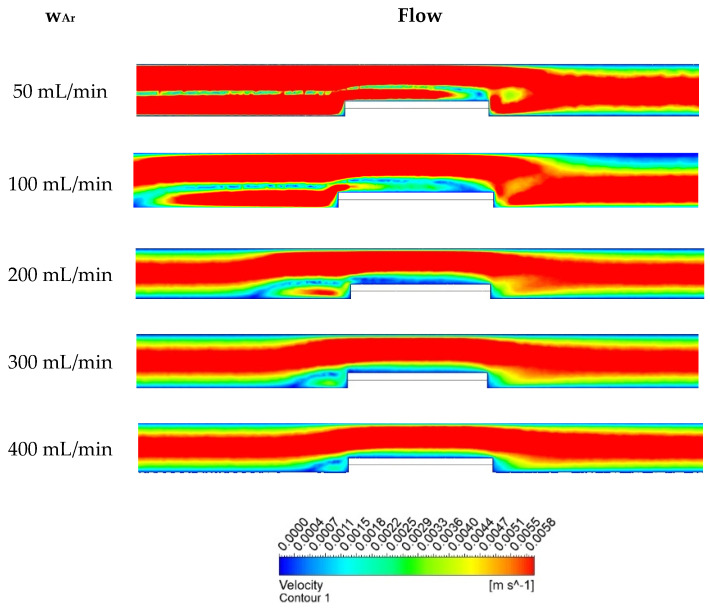
Argon flow character based on flow rate intensity—velocity of argon.

### 3.3. Assessment of the Model Measurement and Physical Measurement

The physical measurements (FM) were compared with the model CFD simulation (MMt—turbulent, MMl—laminar) to determine the deviation according to Equation (11) (maximum value 18%) and the credibility of the software. A correlation between the physical and the model measurement can be observed from Figure 14, Figure 15 and Figure 16.
(11)$$\sigma =\left|\left(\frac{R_{MM}-R_{FM}}{R_{FM}}\right)*100\right|\;(\%)$$
where the following apply:σ—deviation of the result determined as an absolute value; R_MM_—result from the mathematical modelling (CFD); R_FM_—result from the physical measurements.

**Figure 14 materials-17-04627-f014:**
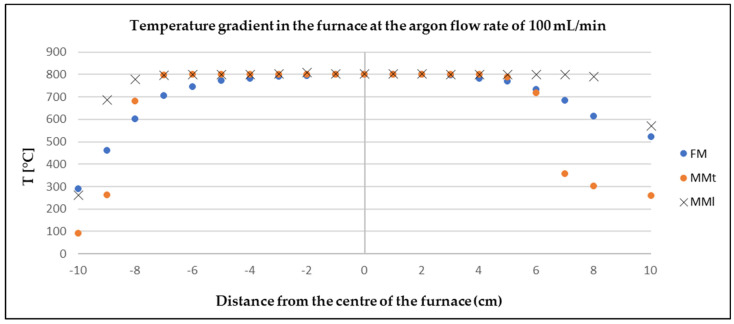
Comparison of model and physical measurements of temperature gradient of argon flow rate of 100 mL/min.

**Figure 15 materials-17-04627-f015:**
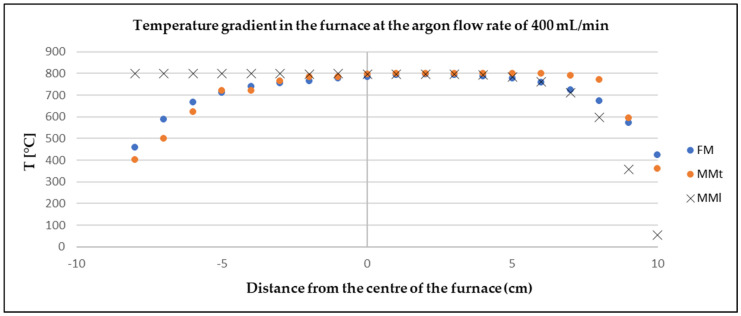
Comparison of model and physical measurements of temperature gradient at argon flow rate of 400 mL/min.

In both cases (FM, MMt), the slight deviations at the outlet and inlet of the furnace were probably due to the actual design of the furnace. When modelling the furnace in Ansys, resistant linings were not modelled. The measured values showed that the argon flow rate had no effect on the temperature in the close vicinity of the zinc sample placed in the centre of the furnace. This finding is very positive as it shows that Ansys software works well and can be used as a reliable tool to predict similar processes.

The results of the measurement of the temperature change in the zinc sample in Figure 16 show that the model measurement was consistent with the physical one. 

**Figure 16 materials-17-04627-f016:**
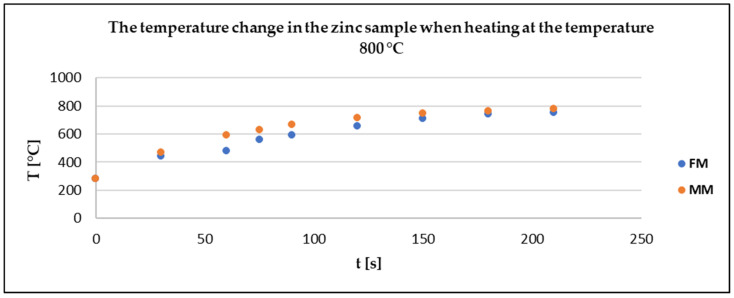
The physical and model measurements of the temperature change in the zinc sample.

Based on the results of the mathematical and physical measurements, the zinc sample reached almost 800 °C after 210 s, which was the set temperature of the furnace. This prediction can provide a better understanding of what time should be considered as the heating time when evaluating the kinetics of the evaporation process. 

After preliminary predictions, the experiments were carried out at 800 °C, with a constant zinc sample dimension (25 × 10 × 5 mm) and different argon flow rates (50–400 mL/min), within 30 min, as described in the following section.

### 3.4. Evaluation of Evaporation Mass Loss Based on Argon Flow Rate

The evaporation of the zinc sample was carried out in a laboratory at argon flow rates of 50, 100, 200, 300, and 400 mL/min. Each flow rate experiment was performed three times, and the average evaporation efficiency was only then calculated. The results of these experiments are shown in Table 6.

Based on these results, it can be concluded that the evaporation efficiency increases with an increasing argon flow rate (Figure 17). It can be concluded that the simulations in Section 3.1 (on the effect of the argon flow rate on the quartz tube temperature) confirmed that the argon flow rate within the observed intensities did not affect the temperature in close proximity to the zinc sample and, hence, they did not affect the evaporation rate itself.

### 3.5. Improvement of the Laboratory Testing by CFD

For verification of the effectiveness of the employment of Ansys, simulations were modelled at flow rates of 500 and 700 mL/min, and a temperature drop around the sample/insert at 700 mL/min was confirmed (Figure 18).

The above simulations show that the argon stream entering at the higher velocity (700 mL/min) and temperature of 25 °C did not have enough time to heat up to the target temperature and, therefore, caused a drop in temperature in the vicinity of the zinc sample. The temperature difference between these two argon flow rates was 9–30 °C in the area of the zinc sample. Considering that the evaporation temperature of zinc is 906 °C and the laboratory experiments were carried out at 800 °C, a further, even small, drop in temperature may have caused a change in weight loss during the evaporation process. This simulation indicated a possible change in the mass loss. The CFD results and the CFD’s prediction of the zinc evaporation loss were compared with the laboratory testing of the zinc evaporation at flow rates of 500 and 700 mL/min, and the results of the zinc mass loss over 30 min are shown in Table 7.

The results in Table 7 show that the increase in the argon flow rate, when comparing the higher flow rates, caused a reduction in the evaporated amount. At an argon flow rate of 700 mL/min, the temperature near the zinc sample dropped and, therefore, the evaporation of zinc was expected to decrease. The CFD modelling proved to be effective for the prediction of such phenomena. 

Additionally, as can be seen in the previous preliminary thermodynamic study by the main author [26], the argon consumption depends on the temperature of the process itself. Based on the results of these two connected studies, it can be concluded that even when the temperature drops due to an argon flow rate increase (which causes a drop in the temperature in an area of a zinc sample), also causing a change in the partial pressure over the zinc sample, the temperature for zinc evaporation must be substituted by the volume of the argon that is put in the system. The same phenomenon can be observed when comparing the evaporation mass losses at 400, 500, and 700 mL/min. The amount evaporated at 700 mL/min was not as high as expected when compared to 500 mL/min, but it still increased when compared to 400 mL/min. However, it can be assumed that at an argon flow rate of 700 mL/min, when the temperature decreases, the main influence on the mass loss is the rate of the vapour movement out of the zinc evaporation zone, and, therefore, the total amount of evaporated zinc is higher. So, when the temperature is lower, the argon consumption must be higher. 

The determination of the evaporation rate dependence at flow rates was useful in terms of the determination of the most suitable argon flow rate, but more valuable information was obtained in the area of post-evaporation products. These findings are discussed in the following section. 

### 3.6. Products Obtained

Based on the conducted experiments, it was found that the argon flow rate influenced the final form of the product obtained, as can be seen in Figure 19 (within the observed argon flow rates of 100–400 mL/min).

The temperature points in the graph in Figure 19 show the temperature in each zone measured physically at each time 1 cm from the centre of the quartz tube placed in the furnace. Based on the measured temperature, the temperature range for the product condensation was assumed.

The graph in the above figure shows that the zinc vapours condensed in the form of metallic zinc at a distance of 4–10 cm from the centre of the furnace. At a distance of 10 cm, the zinc vapours condensed in the form of zinc dendrites and zinc fibres. At lower argon flow rates, the zinc fibres formed prematurely. A zinc “foil” was formed from the zinc vapours at a distance of 10–12 cm from the centre of the furnace, and after this distance, zinc powder was formed. The condensate distribution can also be seen in the photographs in Figure 20.

The change in the intensity of the argon flow rate led to a change in the volume of individual forms of products due to the change in the vapour temperature drop. With the lower argon flow rate, the vapour temperature decreased gradually, while the higher flow rate increased the vapour temperature drop more rapidly, resulting in a different form of condensation. With a lower flow rate, metallic zinc, fibres, and dendrites formed. Zinc powder formed in a higher volume with a higher flow rate (over 200 mL/min). An accurate determination of the mass ratio of the individual products was not possible. The metal droplets remained stuck around the circumference of the quartz tube and, therefore, could not be removed without breaking the tube. The powdered zinc was dispersed along the entire length of the tube, and its fine-grained nature and inaccessible locations in the apparatus made it impossible to obtain the entire quantity. The products obtained can be seen in Figure 21. All the products except for the “film” were observed under a scanning electron microscope (Figure 22) and an EDS diffraction analysis was performed to determine the purity of the products (Figure 23).

From the SEM analysis, it can be concluded that the zinc fibres were of a thickness of approximately 500 nm, and the zinc powder had a spherical morphology with particle sizes of 2–5 μm. Due to the morphology and particle size obtained, these two products are of interest in terms of economic benefits. The implementation of a specific argon flow rate resulted in the production of different forms of zinc condensates based on the temperature gradient.

**Figure 23 materials-17-04627-f023:**
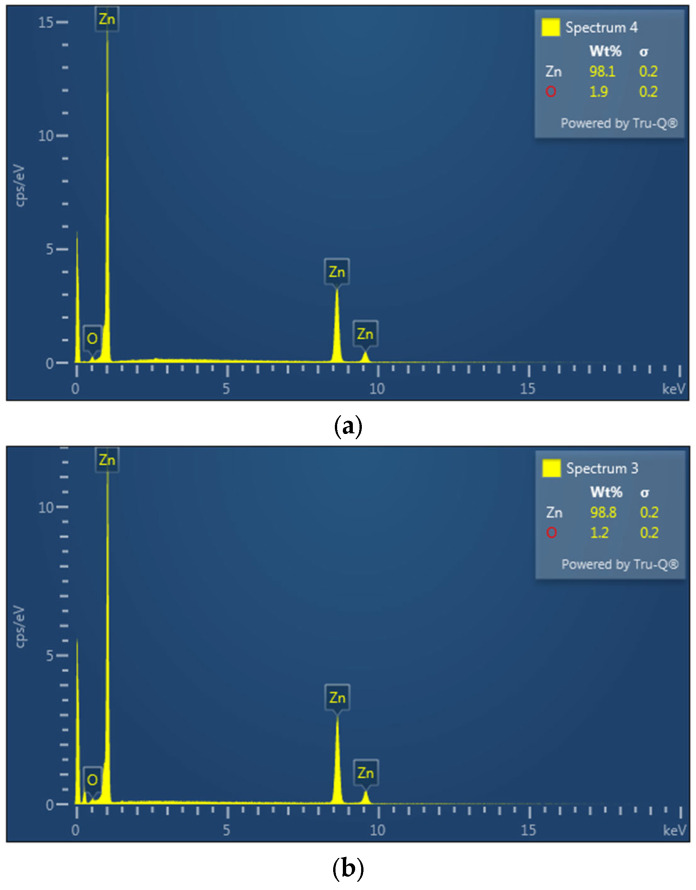
Results of EDS analyses of nanofibres (**a**), powder (**b**), dendrites (**c**), and metallic zinc (**d**).

The EDS analysis proved that the products obtained were of a high purity, with a metallic zinc content over 98%. The lowest purity was assessed to be 98.1% of zinc for the nanofibres, which is logically associated with the biggest specific surface area.

## 4. Conclusions

Based on the results obtained, several statements can be assigned:The observation of the chosen phenomena through the Ansys simulations and physical measurements confirmed correlations with a maximum deviation of 18%. Correlations were observed for phenomena such as heat transfer, temperature gradient, system deflation, and turbulency. These findings are beneficial in terms of the time shortage when adjusting the process conditions, as well as the costs of the overall laboratory experiments conducted.The processing of bottom zinc dross by evaporation at 800 °C with an argon flow rate in the range of 100–400 mL/min resulted in the formation of metallic zinc, dendrite zinc products, and high-added-value nanofibres and powders with particle sizes of 500 nm and 2–5 μm, respectively.Based on the intensity of the argon flow rate (100–400 mL), the compositions of the final products changed. When using an argon flow rate of 100 mL/min, metallic zinc, fibres, and dendrites were produced predominantly. With an argon flow rate over 200 mL/min, zinc powder formation prevailed.

This study showed the option of forming different forms of zinc products when implementing the argon flow rate within these set conditions. This may serve as fundamental to any further investigations in terms of zinc nanofibres or zinc powder production, as a valuable and economically beneficial secondary source of zinc materials. 

Beyond the main objective of the research, which is the process of recycling zinc to obtain high-value-added products, this paper also highlights the possibility of using software to predict some/selected phenomena that can predict certain behaviours during the evaporation process itself. This software has not been used to simulate or predict the actual process of zinc evaporation and condensation. This may be developed as part of future advanced research.

However, the present paper deals only with marginal phenomena, which can help to observe the behaviour of argon when flowing in a quartz tube and in close proximity to a zinc sample. Such initial information/data can predict basic, subsidiary behaviour.

The CFD modelling investigation performed and its limited implementation within the evaporation system showed that there is further need for the mathematical modelling of the zinc evaporation process itself, in terms of vapour transfer from the zinc sample, followed by subsequent condensation. 

## Figures and Tables

**Figure 1 materials-17-04627-f001:**
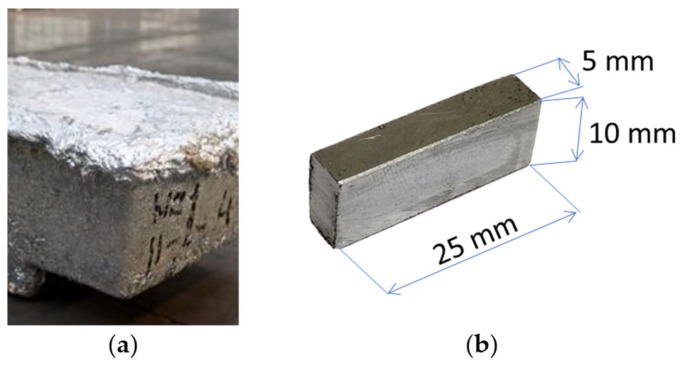
Zinc-bearing waste (bottom zinc dross) (**a**), sample for processing (**b**).

**Figure 2 materials-17-04627-f002:**
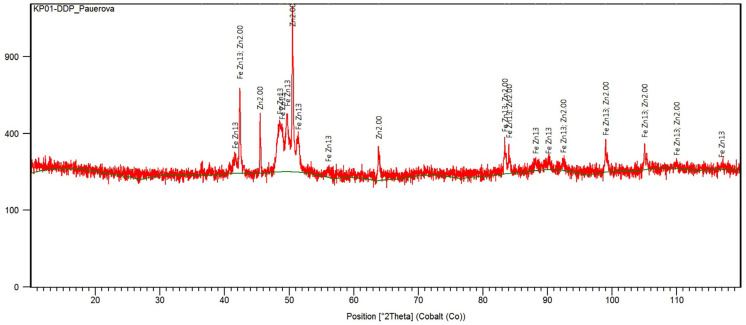
XRD pattern of the bottom zinc dross sample [22].

**Figure 5 materials-17-04627-f005:**
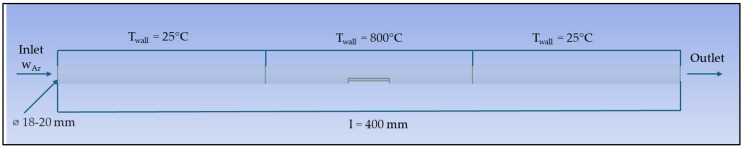
Placement of the sample in the quartz pipe.

**Figure 6 materials-17-04627-f006:**
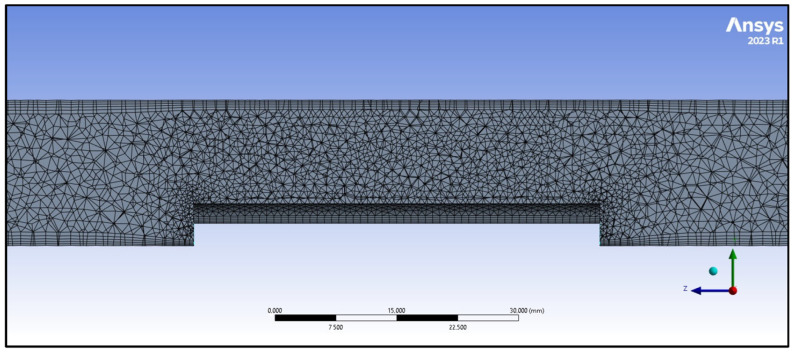
Computational network.

**Figure 7 materials-17-04627-f007:**
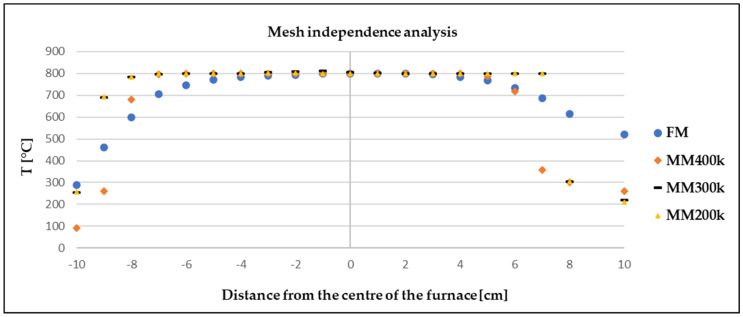
Mesh independence analysis comparing 200–400,000 cells.

**Figure 8 materials-17-04627-f008:**
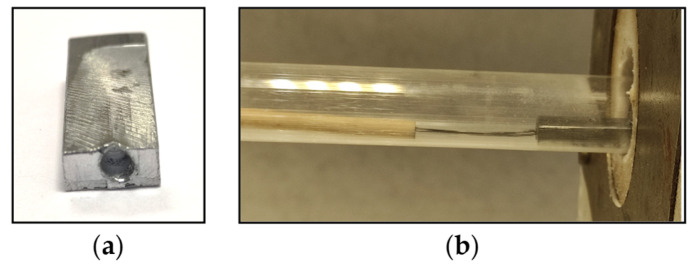
Zinc sample with drilled hole (**a**) and thermocouple inserted into the hole (**b**).

**Figure 9 materials-17-04627-f009:**
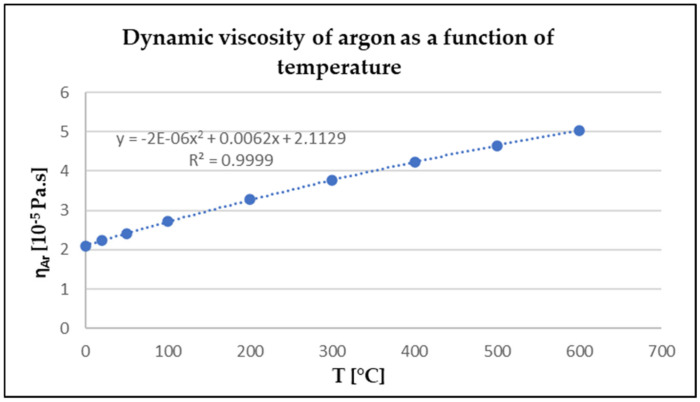
Derivation of the temperature dependence of the dynamic viscosity of argon.

**Figure 17 materials-17-04627-f017:**
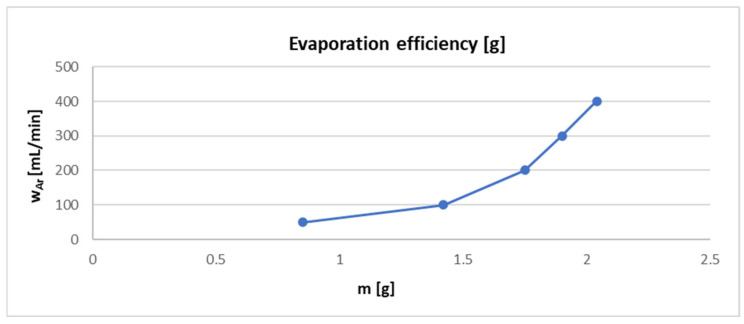
Evaporation efficiency due to the argon flow rate.

**Figure 18 materials-17-04627-f018:**
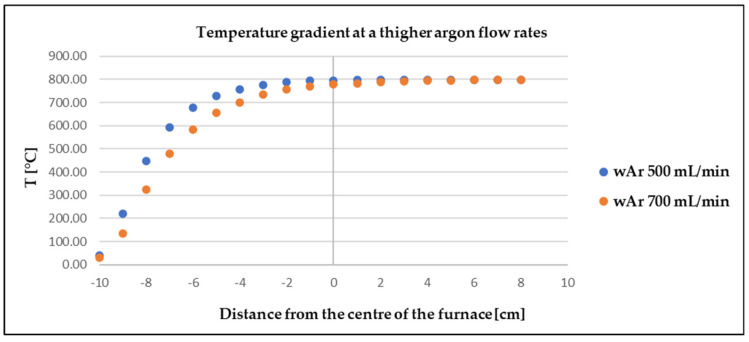
Temperature gradient in proximity of the sample at higher argon flow rates.

**Figure 19 materials-17-04627-f019:**
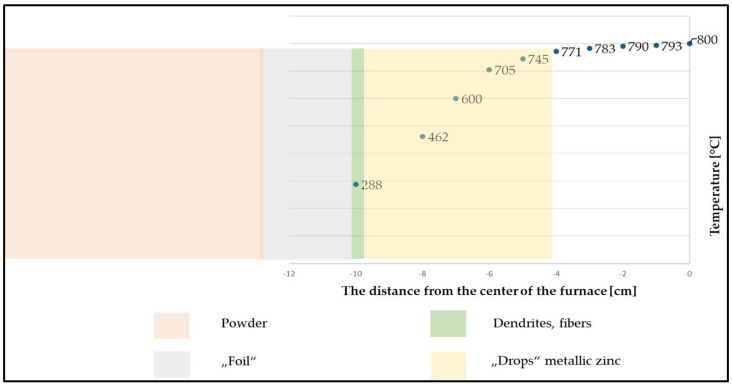
Effect of the temperature gradient in the form of the product obtained—area of products’ condensation after the evaporation.

**Figure 20 materials-17-04627-f020:**
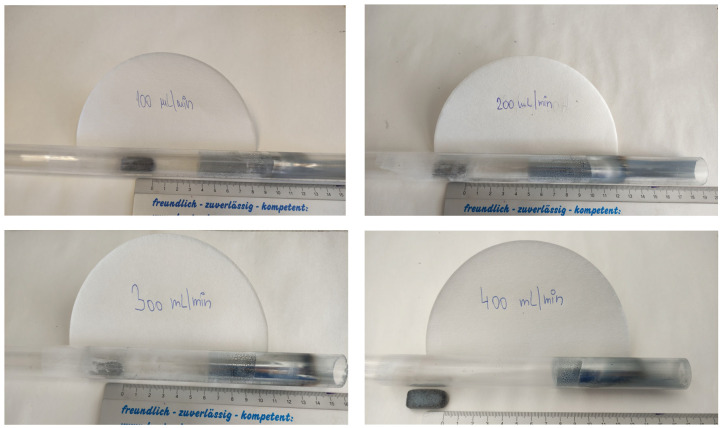
The distribution of the condensates at the observed argon flow rates.

**Figure 21 materials-17-04627-f021:**
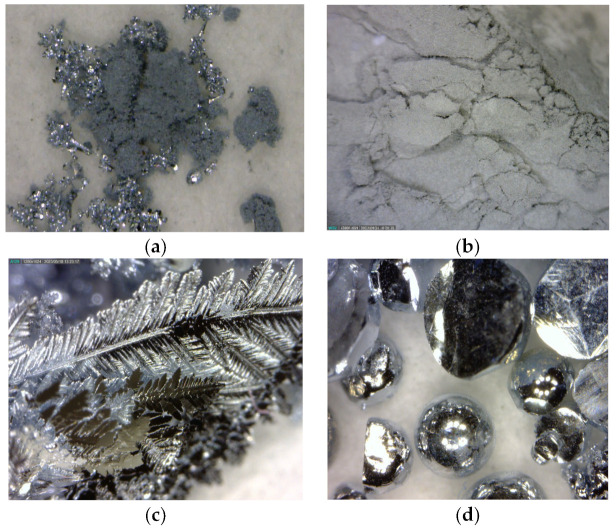
Zinc fibres (**a**), powder (**b**), dendrites (**c**), and metallic drops (**d**) after 30 min of evaporation at argon flow rates of 100 mL/min (**a**,**c**,**d**) and 400 mL/min (**b**).

**Figure 22 materials-17-04627-f022:**
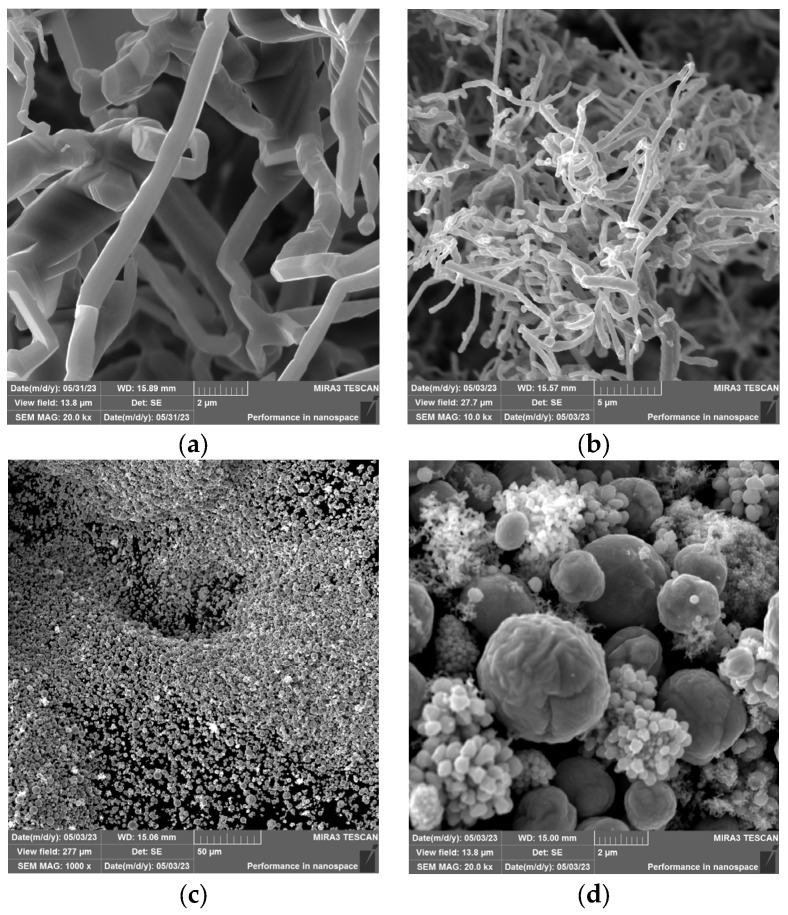
SEM analysis of nanofibres (**a**,**b**), powder (**c**,**d**), dendrites (**e**), and metallic zinc (**f**) introduced in Figure 20.

**Table 1 materials-17-04627-t001:** Chemical analysis of bottom zinc dross sample.

Element	Zn	Fe	Pb	Al	Ni
Amount (wt. %)	94–97	2.2–3.4	0.7–1.5	0.8–1	0.3–0.7

**Table 2 materials-17-04627-t002:** Results of the XRD analysis—phase composition of the sample.

Ref. Code	Compound Name	Chemical Formula
96-901-1600	Zinc	Zn
03-065-1238	Iron Zinc	FeZn_13_
96-900-8478	Lead	Pb

**Table 3 materials-17-04627-t003:** Qualitative values of the skew criterion [24].

Excellent	Very Good	Good	Acceptable	Bad	Unacceptable
0–0.25	0.25–0.50	0.50–0.80	0.80–0.94	0.95–0.97	0.98–1.00

**Table 5 materials-17-04627-t005:** Dynamic viscosity of argon in the temperature range 0–600 °C [25].

T [°C]	0	20	50	100	200	300	400	500	600
η_Ar_ [10^−5^ Pa.s]	2.1	2.23	2.42	2.73	3.28	3.77	4.22	4.64	5.04

**Table 6 materials-17-04627-t006:** Results of evaporation mass loss at chosen argon flow rates after 30 min of evaporation (laboratory testing).

w_Ar_ (mL/min)	Input m (g)	Output m (g)	Mass Loss (g)	x¯ Mass Loss (g)
50	9.17	8.30	0.87	0.85
	9.37	8.46	0.91	
	9.29	8.51	0.78	
100	9.26	7.74	1.52	1.42
	8.11	6.96	1.15	
	9.48	7.88	1.6	
200	9.8	8.19	1.61	1.75
	8.96	6.68	2.28	
	10.19	8.84	1.35	
300	9.38	8.1	1.28	1.90
	9.64	7.66	1.98	
	8.71	6.27	2.44	
400	9.72	7.02	2.7	2.04
	8.75	7.18	1.57	
	8.95	7.11	1.84	

**Table 7 materials-17-04627-t007:** Results of evaporation mass loss average at higher argon flow rates after 30 min of evaporation (laboratory testing).

w_Ar_ (mL/min)	500	700
x¯ mass loss (g)	3.02	2.32

## Data Availability

The original contributions presented in the study are included in the article, further inquiries can be directed to the corresponding author.

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
