# Peer review of "An Experimental Study of Zinc Evaporation from Bottom Zinc Dross at Atmospheric Pressure and in Inert Atmosphere with Integrated CFD Modelling"

_materials, 2024, doi:10.3390/ma17184627_

Round 1
Reviewer 1 Report
Comments and Suggestions for Authors
The article presents laboratory measurements and CFD simulations for the evaporation of zinc dross in argon atmosphere. The article is interesting because it offers valuable laboratory insight in the behaviour of zinc evaporation. However, extensive improvements are necessary in the description of methods, results analysis and discussions. The following points elaborate these issues:
1. Improve the statement of the scientific contribution (lines 138-142). "Simulation of zinc evaporation" is excessively vague. If metal evaporation is available Ansys Fluent, then probably you are not the first to model metal (zinc) evaporation. What aspect/approach of zinc evaporation could be considered as novelty in the present study?
2. How many zinc samples were used in the present study? Table 1 gives the ranges of chemical composition in the zinc dross sample - these ranges were obtained from how many samples?
3. Section 2.3. should give the fundamental equations for mass, momentum and energy conservation, as well as the equations for modeling zinc evaporation. Also, the boundary conditions should be stated (tube wall, argon outflow, zinc/argon contact surface). The entire quartz tube was simulated or axisymmetry was used? What is the length of the quart tube? What is the argon inlet temperature? What is the heat transfer between the quartz tube and the surroundings? What is the time step of the simulation, what are the number of iterations per time step and the convergence criteria?
4. The explanation concerning figure 7 in section 2.3.2 should be improved. What are the differences between the five temperature contours in figure 7 and why?
5. Section 2.3.4. what is the Reynolds number for argon flow in the tube? What is shown in figure 10 - velocity contours? Likely the flow was not turbulent but rather unsteady laminar if Re < 2300. What flow model was used in Ansys Fluent - laminar of turbulent?
6. Figures 11 and 12: what is the meaning of the labels "FM" and "MM"? Blue dots are for measurements while orange dots are for CFD? How did you measure the temperature in the furnace - what type of temperature sensor was used and how many measurement points? In figure 11, three MM points (at 7-10 cm) are significantly lower than the corresponding FM points - what causes this discrepancy?
7. Figure 13 shows the temperature in the zinc sample, not the heat transfer. Figure 9 shows the location of the temperature measurement on the zinc sample, but where is this point in the CFD domain (figure 5)?
8. Table 3: the data is from laboratory measurements or CFD simulations? The "evaporation efficiency" should be simply termed "evaporation mass loss".
9. Figure 15 is unclear, what is the meaning of the temperature points? This figure shows the final composition after evaporation? What are the mass fractions of powder, foil, dendrites and metallic drops in the final composition? How does the temperature affect the final composition of the product?
10. What was the intent of using CFD simulations? Beside obtaining temperature and flow fields, did the CFD generate any results concerning the evaporation rate that could be compared to the measurements in section 3.2.? How did CFD improve the laboratory testing of zinc evaporation?
11. The conclusions should also summarize the principal limitations in the methods of the present study and suggest guidelines for the improvement in future research.
Comments on the Quality of English LanguageThe writing style should be improved by a native English speaker or an English editing office.
Author Response
Hello dear Mr. or Mrs., please see the attachment.

Reviewer 2 Report
Comments and Suggestions for Authors
The article describes model and experimental studies of the zinc evaporation process from a waste sample. The article combines CFD model studies and laboratory measurements, which, as always, is a valuable contribution to the development of cognitive knowledge. The article may be of interest to both research teams and technological process developers. The reviewer presents his comments on the article below, which should be explained and supplemented.
1, Introduction - the literature references are somewhat old, the latest articles should be supplemented. Bulk citations [10-17] are not elegant. In the introduction, special attention should be paid to the application aspect of this process, also taking into account economic conditions.
2. All figures should be reviewed, paying attention to their substantive content and form. Figures should be combined and the number of figures should be limited to those that are essential. For example, figures 3-4 provide little information, lack dimensioning. In many figures, the font is too small and has different styles. Fig. 12-13, the axis is temperature, not gradient or heat transfer, one figure has a title at the top, the other does not, the physical measurement has the abbreviation FM. In some figures, the captions overlap or go beyond the figure. The figures are not prepared for publication!
3. No precise data of the physical experiment. They should be provided so that other teams can verify the measurement. Type, drawing of the furnace chamber, precise data of thermometers and their location, measurement uncertainties, etc. etc.
4. The modeling concerns a simple flow and heat transfer phenomenon and has little connection with evaporation. What is the authors' novel and original contribution here?
5. Do photos 17-18 come from the studied process? If so, they should be linked to the process parameters.
6. The conclusions should be reworded to present the original and innovative achievements of this research.
Proofreading by a native speaker with knowledge of the scientific subject is necessary
Author Response

(The authors gave the same response as above.)

Reviewer 3 Report
Comments and Suggestions for Authors
The manuscript deals with an experimental and numerical analysis of Zinc evaporation from bottom Zinc Dross. There are a lot of issues that need to be solved by the authors.
- The following questions must be answered in the abstract section: What and how was it done? Why was the study done? What was found? What do these findings mean and what impact do they have? In this sense, it is not clear the purpose (why was done?) as well as the novelty and impact of the study.
- Second keyword is not relevant.
- Lines 94-105. It seems to me that the introduction to CFD tool showed in these lines, must be shorter and concise. Therefore, it will not require several references.
- Ensure the references are cited using the proper format.
- The goal of the introduction section is to demonstrate to the reader the need of the present investigation, however, this is not the case. Expand and explain in a better form the previous studies and highlight the differences between those and the present one.
- Discuss and explain figure 2. Does figure 2 include a table? Why is it not Table 2?
- Line 163. Justify these values.
- Figure 4. Add dimensions to the geometry.
- Line 173. The Navier-Stokes equations are not the same as the Euler equations. Which ones were used? Please add them to the manuscript. Additionally, what assumptions were taken into account?
- Table 2 needs a reference.
- Mesh Independence analysis is missing.
- Improve the quality of figure 5.
- Add boundary conditions, Values of those conditions, Models used, solution algorithms, etc.
- Line 194. Justify these values.
- Explain Figure 7, as it shows practically the same behavior. What is the conclusion of this figure? What can be inferred from it? Is it necessary to have such high precision on the temperature scale?
- Line 206. Why inventor and not Fluent? What is the explanation for using 2 different software?
- Lines 230-231. But if the calculations indicate laminar flow, why does it exhibit turbulent behavior? Did the simulation account for a turbulence model, and if so, which one?
- The results of the simulation must be in section 3 and not in section 2.
- Section 3.1. If this is the validation of the numerical model, it should be presented before the simulation results, not after.
- Line 242. How was this deviation calculated?
- Figure 11. What is FM and MM? Also, nomenclature section is missing.
- Some references are outdated and should be replaced with more current ones.
- In general, the manuscript is poorly organized and lacks important details for proper understanding, particularly in the simulation section. A correct justification of many of the values used, as well as the various assumptions made, is required. Additionally, it is essential to highlight the novelty of this manuscript and clearly define the knowledge gap it aims to fill.
Author Response

(The authors gave the same response as above.)

Round 2
Reviewer 1 Report
Comments and Suggestions for Authors
Following the first round of revisions, the article has been improved. However, there are still several important issues that must be resolved:
1. Section 2.3.: it is stated that the flow is laminar (Re < 2300Ž) and in all instances the Reynolds number is smaller than 50 (table 4). Surprisingly, the k-e turbulence model is used. What is the reasoning for using the k-e turbulence model at such low Reynolds numbers, what references from the literature could support this decision? Also, beside equations (1) and (2), conservation equations for mass, momentum and energy should also be given in this section.
2. Figures 13-15 compare measurements (FM) with CFD simulations (MM). How can we know that the k-e model produces more accurate results than the laminar model if the results of the laminar model are not shown? Figures 13–15 should also include results from the laminar model. Line 327: "CFD measurement" should be replaced with "CFD simulation".
3. The boundary conditions described in line 228-235 should also be marked in figure 5 and/or figure 6. What is the inlet pressure of the argon stream?
4. Table 4: The velocity at 25 °C for 500 mL/min should be 3.27 e-2 m/s, not 3.10 e-2 m/s.
5. Equations (3) and (4): "kin.vis" and "dyn.vis" should be replaced with the standard symbols for kinematic and dynamic viscosity, respectively.
6. The dynamic viscosity of argon at 25 °C in table 4 does not agree with the dynamic viscosity given in table 5. The dynamic viscosity at 25 °C should be 2.26e-5 Pa×s. The dynamic viscosity at 800 °C is 5.84e-5 Pa×s, according to REFPROP and COOLPROP data. Table 5, equations (6) and (7), and figure 8 are not really necessary, just a reference to a database for fluids physical properties.
7. This sentence makes no sense: "The CFD results were compared with laboratory testing and the results are shown in Table 7." Table 7 reports results from laboratory measurements; no CFD results are given in Table 7.
8. Figure 16: how much is the difference in the temperature drop in the vicinity of the zinc sample between a flow rate of 500 mL/min and a flow rate of 700 mL/min? The two temperature contours shown in figure 16 are very much alike.
9. Lines 369-371: you state that "for an argon flow rate of 700 mL/min, the temperature near the zinc sample dropped and therefore the evaporation of zinc was expected to decrease." However, table 7 shows that the evaporative mass loss increases with the flow rate, and for a flow rate of 700 mL/min, the evaporation mass loss is the greatest (average 2.04 g after 30 minutes). You need to clearly explain how and why does the argon flow rate and zinc sample temperature affect the evaporation mass flow?
Comments on the Quality of English LanguageModerate editing is necessary. The writing style needs to be improved. The text should be checked for typos, and non-standard symbols should be replaced with the standard ones.
Author Response
Dear Reviewer, please see the attached document.

Reviewer 2 Report
Comments and Suggestions for Authors
The authors have made changes and additions to the text in accordance with the reviewer's suggestions. The article currently meets the requirements of a publication in terms of scientific content. After a thorough review of the article and the introduction of formal corrections regarding formatting and correction of errors, the article can be published.
Comments on the Quality of English LanguageEditing by a native speaker will certainly add fluidity to the content of the article.
Author Response
Dear Mr. or Mrs.,
Thank you for your help.
Best Regards,
Pauerová Katarína
Reviewer 3 Report
Comments and Suggestions for Authors
The authors did not correctly resolve the problems in the manuscript.
- The impact and novelty in the abstract are still not highlighted.
- "Error reference source not found" appears in the manuscript.
- A mesh independence analysis is still needed.
- Better justification of the turbulence model is needed. Add the values of the constants.
- The same analysis can be done in Fluent; I do not see the advantage of using two software programs.
- The nomenclature section is still missing.
Author Response
Dear reviewer, please see the attached document.

Round 3
Reviewer 1 Report
Comments and Suggestions for Authors
After two rounds of revisions, my opinion is that the article has been improved enough to warrant publication.
Comments on the Quality of English LanguageMinor editing necessary.
Author Response
Thank you for your comments
Reviewer 3 Report
Comments and Suggestions for Authors
Some references are outdated and should be replaced with more current ones. It is a critical issue that needs to be addressed to ensure the manuscript reflects the most up-to-date research in the field.
Author Response
Some references are outdated and should be replaced with more current ones. It is a critical issue that needs to be addressed to ensure the manuscript reflects the most up-to-date research in the field.
Thank you for your suggestion. The references in the article have been updated. Reference number 1 was left due to the fact that one of the starting studies in which argon was used as an inert gas.